# The Silent Conversation: How Small RNAs Shape Plant–Microbe Relationships

**DOI:** 10.3390/ijms26062631

**Published:** 2025-03-14

**Authors:** Jie Liu, Yuntong Lu, Xiaoyan Chen, Xing Liu, Yunying Gu, Fei Li

**Affiliations:** 1School of Life Sciences, Guizhou Normal University, Guiyang 550025, China; liujie791204@126.com (J.L.); 18984028920@163.com (Y.L.); 222100100431@gznu.edu.cn (X.L.); 15718669671@163.com (Y.G.); 2Key Laboratory for Information System of Mountainous Area and Protection of Ecological Environment of Guizhou Province, Guizhou Normal University, Guiyang 550025, China; chenxy8387@126.com; 3State Key Laboratory of Microbial Technology, Shandong University, Jinan 250016, China

**Keywords:** cross-kingdom RNA interference (ckRNAi), small RNAs (sRNAs), plant–microbe interactions, host-induced gene silencing (HIGS), extracellular vesicles (EVs)

## Abstract

This review highlights the emerging role of cross-kingdom RNA interference in plant–microbe interactions, particularly the transfer of sRNAs from microbes to plants and vice versa, emphasizing the importance of this mechanism in both mutualistic and pathogenic contexts. As plants adapted to terrestrial life, they formed symbiotic relationships with microbes, essential for nutrient uptake and defense. Emerging evidence underscores sRNAs, including small interfering RNAs (siRNAs) and microRNAs (miRNAs), as critical regulators of gene expression and immune responses in plant–microbe interactions. In mutualistic symbioses, such as mycorrhizal fungi and nitrogen-fixing bacteria associations, sRNAs are hypothesized to regulate nutrient exchange and symbiotic stability. In pathogenic scenarios, microbes utilize sRNAs to undermine plant defenses, while plants employ strategies like host-induced gene silencing (HIGS) to counteract these threats. We further explore the emerging role of extracellular vesicles (EVs) in sRNA transport, which is critical for facilitating interspecies communication in both pathogenic and mutualistic contexts. Although the potential of ckRNAi in mutualistic interactions is promising, the review highlights the need for further experimental validation to establish its true significance in these relationships. By synthesizing current research, this review highlights the intricate molecular dialogues mediated by sRNAs in plant–microbe interactions and identifies critical gaps, proposing future research directions aimed at harnessing these mechanisms for agricultural advancements.

## 1. Introduction

The migration of plants onto land exposed them to a variety of microbes, shaping both mutualistic and pathogenic interactions that influenced plant evolution [1]. Early plants likely relied on symbiosis with microbes, such as arbuscular mycorrhizal (AM) fungi, which facilitated nutrient uptake and pathogen defense [2]. In return, obligate biotrophic AM fungi rely entirely on carbon from the host plant, mainly in the form of fatty acids [3,4]. During symbiosis, the fungi invade plant roots, forming branched arbuscules within cortical cells, while the plant plasma membrane envelops the fungal hyphal structures, creating the peri-arbuscular membrane (PAM) and facilitating the exchange of nutrients and signaling molecules, including small RNAs (sRNAs) [3].

Recent studies have highlighted the critical role of sRNAs in regulating plant responses, especially in plant–microbe interactions. These sRNAs, which include small interfering RNAs (siRNAs) and microRNAs (miRNAs), are crucial in the regulation of gene expression and immune responses in plants [5]. Plant sRNAs mediate RNA interference (RNAi) through various mechanisms, targeting both endogenous and foreign RNA sequences to silence genes. The RNAi machinery involves Dicer-like (DCL) proteins processing precursor sRNAs, which are then loaded onto ARGONAUTE (AGO) proteins to mediate gene silencing via RNA cleavage or translational repression [6]. Interestingly, some of these sRNAs have been shown to act in cross-kingdom RNA interference (ckRNAi), where sRNAs are exchanged between the host and microbial partners, influencing gene expression across species boundaries. This mechanism regulates both pathogen defense and mutualistic plant–microbe interactions [7,8].

The concept of cross-kingdom RNAi (ckRNAi) has been further explored in both pathogenic and mutualistic symbioses, providing a framework for understanding how plants and microbes communicate at the molecular level. For example, in pathogenic interactions, microbes like fungi and bacteria utilize sRNAs to target host immunity genes, such as those involved in the MAPK cascade, receptor kinases, and transcription factors [9]. These pathogen-derived sRNAs can impair plant immunity by silencing defense-related genes, thus promoting infection [10]. Conversely, plants also employ sRNAs in a form of defense against pathogens, targeting microbial genes through processes like host-induced gene silencing (HIGS) [11].

In the context of mutualistic symbioses, such as AM symbiosis, the role of ckRNAi remains an emerging area of study. Evidence suggests that, in these relationships, microbes such as AM fungi may use sRNAs to manipulate plant gene expression, particularly in genes related to defense and membrane remodeling during the formation of the PAM [12]. Furthermore, plant-derived sRNAs have been proposed to regulate microbial function in symbiosis, although experimental validation of this in the AM context is still underway [13]. These findings underscore the bidirectional nature of sRNA-mediated regulation in plant–microbe interactions, with sRNAs acting as crucial mediators of both symbiosis and immunity.

This review aims to summarize the current understanding of sRNA involvement in plant–microbe interactions, focusing on cross-kingdom RNA interference. We examine how these mechanisms influence both pathogenic and mutualistic symbioses, with an emphasis on the potential for RNAi to regulate plant–microbe communication in AM symbiosis and other mutualistic systems.

## 2. Exploring Pathogen–Plant Communication: The Mechanisms and Challenges of ckRNAi

### 2.1. Pathogen Strategies: Exploiting Cross-Kingdom RNA Interference to Modulate Plant Immunity

Recent research has demonstrated that certain pathogens, including fungal and heterokont species, employ cross-kingdom RNA interference (ckRNAi) to regulate plant immune responses. Through the prediction of complementary sRNA–mRNA sequences and subsequent cleavage assays, it has been shown that pathogens target key immune response genes in plants, including those involved in the mitogen-activated protein kinase (MAPK) cascade, such as *MPK1*, and various receptor-like kinases (LRR-RKs) and transcription factors like FEI2 and WRKY7 [10,14]. In some cases, a single sRNA may target multiple host genes, as observed with *Botrytis cinerea* siR37, which simultaneously targets genes such as *FEI2*, *WRKY* transcription factors, and defensins in *Arabidopsis thaliana*, thereby increasing the plant’s susceptibility to infection [9].

Pathogen-derived sRNAs involved in ckRNAi often interact with the host’s ARGONAUTE1 (AGO1) protein. This interaction, characterized by a uridine-rich bias in the 5’ end of the sRNA, facilitates the silencing of host transcripts through post-transcriptional gene regulation [15]. The importance of AGO1 in this process is further evidenced by the increased resistance to *B. cinerea* seen in *Arabidopsis ago1* mutants [10]. However, there is growing evidence that some pathogens may manipulate different plant AGOs in ckRNAi, as shown in the case of *Fusarium oxysporum* f. sp. *lycopersici* and its 23 nt miRNA-like molecule Fol-milR1, which preferentially binds AGO4a in *Solanum lycopersicum* to suppress the expression of a host defense-related calcium-binding protein kinase [16]. AGO4, which typically functions in RNA-directed DNA methylation, appears to facilitate silencing of plant genes through both transcriptional and post-transcriptional pathways, adding complexity to the understanding of pathogen-mediated gene regulation [15].

Despite early groundbreaking studies by Weiberg et al. on plant–microbe ckRNAi, recent experiments have challenged the significance of this mechanism. In a study on *S. lycopersicum* and *B. cinerea*, the knockout of key fungal Dicer-like proteins (BcDCL1 and BcDCL2) led to a reduction in predicted ckRNAi-acting sRNAs but did not affect fungal virulence, suggesting that ckRNAi may not play a crucial role in this host–pathogen interaction [17]. This conclusion, however, has been contested, with the rebuttal arguing that experimental conditions—such as the use of an unstable ku70 background—rendered the results inconclusive [18,19]. Additionally, inconsistencies in bioinformatics methods and a lack of experimental validation regarding the cleavage of host transcripts by *B. cinerea* sRNAs further question the validity of the findings. These challenges highlight the need for rigorous experimental validation to confirm or refute the role of ckRNAi in plant–microbe interactions.

Moreover, recent studies have suggested that sRNAs may not be the only type of RNA involved in plant–pathogen communication. For instance, *Ustilago maydis*, a fungal pathogen of maize, has been shown to transfer mRNA species associated with extracellular vesicles (EVs) into host cells. These mRNAs, enriched for genes involved in nitrogen and glycerolipid metabolism, as well as aromatic amino acid biosynthesis, suggest that pathogens may reprogram host metabolism by transferring metabolic enzymes directly into the plant [20,21]. This mechanism may provide a direct advantage to the pathogen by facilitating the invasion of host tissue, though it has not yet been observed in other plant–pathogen systems. It is possible that this mechanism is specific to *U. maydis*, which lacks the RNAi machinery necessary for manipulating host gene expression through ckRNAi [22].

In conclusion, while cross-kingdom RNA interference (ckRNAi) presents a promising avenue for understanding plant–pathogen communication, its full significance remains to be validated. The complexity of host–pathogen interactions, coupled with the involvement of various plant ARGONAUTE (AGO) proteins, underscores the intricate regulatory networks governing plant immune responses. Furthermore, the discovery of alternative RNA transfer mechanisms employed by certain pathogens introduces additional layers of complexity, highlighting the need for further exploration into the diverse ways sRNAs can modulate plant–microbe interactions. Recent experimental challenges also emphasize the critical role of rigorous and refined methodologies in deciphering the nuanced functions of sRNA-mediated communication. To fully appreciate the multifaceted roles of sRNAs, particularly in the context of both pathogenic and mutualistic symbioses, it is essential to continue unraveling the mechanisms underlying their transport, uptake, and regulation. This deeper understanding will enrich our knowledge of how sRNAs influence plant immunity, pathogen virulence, and symbiotic relationships, offering new insights into the dynamic interplay between plants and their microbial partners.

### 2.2. Plant-Offensive: Utilizing Cross-Kingdom RNA Interference to Target Pathogen Virulence

The study of plant-to-pathogen small RNA (sRNA) transport and cross-kingdom RNA interference (ckRNAi) is a relatively new area of research, though examples of this phenomenon have been documented for both fungal and oomycete pathogens (Table 1; Figure 1). Much like the reciprocal event of ckRNAi, plant-driven ckRNAi plays a crucial role in modulating gene expression during infection. Specifically, plants may target genes essential for the pathogen’s survival and reproduction. For instance, in the *Arabidopsis*–*Botrytis cinerea* system, the plant targets genes associated with virulence, such as components of the secretion system [23]. Similarly, in the *Arabidopsis*–*Phytophthora capsici* interaction, host sRNAs regulate splicing factors crucial for pathogen reproduction [24].

A surprising development in recent studies of host-induced gene silencing (HIGS) is the emerging evidence that plants may extend this mechanism beyond fungal pathogens to target bacterial pathogens, such as *Pseudomonas syringae*, despite the absence of a conventional RNA interference (RNAi) pathway in bacteria [25]. This raises intriguing questions about the underlying mechanisms enabling cross-kingdom RNAi (ckRNAi) in bacterial systems. One proposed explanation is that plant-derived small RNAs (sRNAs) could be internalized by bacterial cells through yet unidentified uptake pathways, potentially involving extracellular vesicles or other RNA transport mechanisms. Additionally, recent findings suggest that plant AGO1 proteins, which are central to RNAi processes in eukaryotic cells, may play a crucial role in facilitating this interaction. Specifically, AGO1 has been shown to bind extracellular sRNAs that participate in ckRNAi, and it is hypothesized that this binding could enhance the stability and targeted delivery of these sRNAs to bacterial cells, allowing them to exert gene regulatory effects within the pathogen [26]. These discoveries challenge the conventional understanding of HIGS and suggest a broader role for sRNAs in plant immunity, warranting further investigation into the mechanisms and biological significance of RNA-based plant defense strategies against bacterial pathogens. This discovery presents the intriguing possibility that plants could use their native sRNAs in ckRNAi as a defense against a wide variety of pathogens, adding a new layer to our understanding of plant immunity. Further research into this aspect of plant–pathogen communication is warranted.

**Table 1 ijms-26-02631-t001:** Experimentally verified studies on ckRNAi.

Host Species	Parasite/Mutualist Species	Type of Interaction	Target Gene(s) or Gene Ontology (GO) Terms	sRNA	Reference
*Glycine max*	*Bradyrhizobium japonicum*	Mutualistic	ROOTHAIRDEFECTIVE3 (RHD3a/RHD3b) HAIRYMERISTEM4 (HAM4a/HAM4b) LEUCINE-RICHREPEAT EXTENSION-LIKE5 (LRX5)	21 nt tRNA fragments (tRFs) Bj-tRF001, Bj-tRF002, Bj-tRF003	[7]
*Arabidopsis thaliana*	*Botrytis cinerea*	Pathogenic	PRXIIF (peroxiredoxin) MPK1 MPK2 Wall-associated kinase	21 nt sRNAs Bc-siR3.1, Bc-siR3.2, Bc-siR5	[10]
*Solanum lycopersicum*	*Botrytis cinerea*	Pathogenic	MAPKKK4 (Mitogen-activated protein kinase kinase kinase)	21 nt sRNA Bc-siR5	[10]
*Arabidopsis thaliana*	*Botrytis cinerea*	Pathogenic	WRK7 WRKY57 FEI2 (LRR-RK) PMR6 (pectinlyase) ATG5 (defensin)	Bc-siR37	[9]
*Triticum aestivum*	*Puccinia striiformis* f. sp. *tritici*	Pathogenic	Pathogenesis-related2 (PR2) gene SM638 (b1,3glucanase)	miRNA-like (milR1)	[9]
*Arabidopsis thaliana*	*Cuscuta campestris*	Pathogenic	TIR1 AFB2, AFB3, BIK1 SEOR1 (phloem protein) HSFB4 (transcriptional repressor)	22 nt miRNAs e.g., miR393	[27]
*Arabidopsis thaliana*	*Sclerotinia sclerotiorum*	Pathogenic	SNAK2 (SNF1-related kinase) SERK2 (somaticembryogenesis receptor-like kinase2)	22–23 nt TE-derived sRNAs	[28]
*Solanum lycopersicum*	*Fusarium oxysporum* f. sp. *lycopersici*	Pathogenic	FRG4 (calcineurinB-like-interacting protein kinase)	23 nt miRNA-like Fol-milR1	[16]
*Triticum aestivum*	*Puccinia striiformis* f. sp. *tritici*	Pathogenic	19 target genes including: TraesCS2D02G510300.1 (NB LRR) TraesCS3A02G302100.1 (glutathione S-transferase) TraesCS7B02G299200.1, TraesCS4D02G316900.1 (bZIP transcription factors)	1720–21 nt sRNAs	[29]
*Malus x domestica*	*Valsa mali*	Pathogenic	RLKT1, RLKT2 (receptor-like protein kinases involved in defence signaling)	miRNA-like Vm-milR1	[30]
*Solanum lycopersicum*	*Botrytis cinerea*	Pathogenic	ATG2 (Autophagy-related2) MPKKK4 (Mitogen-activated protein kinase kinase kinase) PPR (Pentatrico peptide repeat protein) ACIF1 (Avr9/Cf-9–INDUCEDF BOX1)	21 nt sRNAs Bc-siR3.1, Bc-siR3.2, Bc-siR5	[19]
*Oryza sativa*	*Xanthomonas oryzae* pv. *oryzicola*	Pathogenic	JMT1 (Jasmonate methyltransferase)	Xosr001	[31]
*Gossypium hirsutum*	*Verticillium dahliae*	Pathogenic	Ca2+ dependent cysteine protease (Clp-1), isotrichodermin C-15 hydroxylase (HiC-15)	miR166, miR159	[32]
*Arabidopsis thaliana*	*Botrytis cinerea*	Pathogenic	BC1G_10728–Vps51 (Vacuolar protein sorting 51) BC1G_10508–DCTN1 (dynactin subunit) BC1G_08464-Suppressor of Actin (SAC1)-like phosphoinositide phosphatase	TAS1c-siR483, TAS2-siR453	[23]
*Triticum aestivum*	*Puccinia striiformis* f. sp. *tritici*	Pathogenic	9 target transcripts including: KNF02052, KNF02053 (glycosyl hydrolase family 26), KNE96707 (60S ribosomal protein L11)	818–24 nt sRNAs	[29]
*Arabidopsis thaliana*	*Verticillium dahliae*	Pathogenic	Ca2+ dependent cysteine protease (Clp-1), isotrichodermin C-15 hydroxylase (HiC-15)	miR166, miR159	[33]

### 2.3. Extracellular Vesicles in Plant–Microbe Dialogues: Mechanisms and Controversies

The topic of extracellular vesicles (EVs) has been extensively discussed, particularly regarding their role as products of non-canonical secretion mechanisms like multivesicular bodies (MVBs) or exocyst-positive organelles (EXPOs). These vesicles are recognized for their significant role in microbial interactions with animal hosts by transporting vital toxins and infectious proteins [33]. Recent studies also indicate that EVs play a similar role in plant–microbe interactions, facilitating infection in scenarios that are pathogenic [23] as well as mutualistic, such as in the root-nodulating symbiosis with *Sinorhizobium fredii* [34]. In such mutualistic interactions, there is a notable downregulation of defense genes and an upregulation of symbiosis-specific transcription factors, although the exact mechanisms remain unspecified. This observation aligns with evidence from other root-nodulating symbioses that suggest an EV-dependent mechanism for sRNA transport [7]. While the mechanisms for EV cargo uptake are still not thoroughly understood, recent findings imply that clathrin-mediated endocytosis (CME) might be involved in the uptake of fungal EVs by plant hosts in systems like *Arabidopsis*–*B. cinerea* [19]. However, it is yet to be determined whether sRNAs are primarily encapsulated within EVs or found outside them. RNase protection assays suggest encapsulated sRNAs in *Arabidopsis–B. cinerea* pathosystems [23], although stringent assays also indicate a predominant association of sRNAs with RNA-binding proteins (RBPs) in the extracellular space [35]. Despite this, RBPs may form part of an EV protein corona, observed in mammalian EVs but not yet described in plants [36]. Similar RBP-mediated transport mechanisms are noted in organisms like nematode worms, suggesting this as a valid sRNA export method in ckRNAi [37]. Extracellular vesicles (EVs) have emerged as significant players in the interactions between arbuscular mycorrhizal (AM) fungi and plants [13]. In the context of AM symbiosis, EVs have been proposed to mediate communication between the fungal symbiont and its plant host, potentially facilitating the transfer of small RNAs (sRNAs) or other bioactive molecules that influence host gene expression and contribute to symbiotic establishment [13]. Similar to other plant–microbe interactions, EVs derived from AM fungi may be involved in regulating plant defense mechanisms and promoting the upregulation of genes associated with symbiosis. For example, studies have shown that EVs released by AM fungi could be responsible for the transport of fungal-derived sRNAs, which might modulate plant immune responses and enhance nutrient exchange between the two organisms. However, the precise mechanisms by which these vesicles are taken up by plant cells remain unclear. The role of EVs in sRNA transport remains a subject for further investigation, particularly in PAS systems where EVs are proposed sRNA carriers [38]. This role is supported by TEM evidence of MVBs fusing with PAM and implicated in sRNA delivery from plants to pathogens [23]. More rigorous studies will be necessary to identify sRNAs and their precise localization concerning EVs [36,39].

## 3. The Role of sRNAs in Mutualistic Symbiosis

### 3.1. Microbe-to-Plant Communication in Mutualistic Interactions

The exploration of small RNA (sRNA) transport and cross-kingdom RNA interference (ckRNAi) in mutualistic plant–microbe interactions is still in its early stages. However, emerging evidence indicates that such interactions are crucial in symbioses, such as the one between *Medicago truncatula* and the nitrogen-fixing bacterium *Bradyrhizobium japonicum*. In this symbiotic relationship, tRNA-derived fragments (tRFs) are produced by the bacteria and contribute to the regulation of host gene expression, particularly genes involved in root development essential for nodule formation. Notably, tRFs have been shown to target key genes like *ROOT HAIR DEFECTIVE 3* and *HAIRY MERISTEM 4*, and when these tRFs are mimicked, nodule formation and bacterial colonization are significantly reduced [7]. Furthermore, rhizobial tRFs have been found to interact with *Glycine max* AGO1, and a similar mechanism is observed in the *Phaseolus vulgaris*–*Rhizobium tropici* symbiosis, indicating that host RNAi can be manipulated in a manner similar to ckRNAi seen in plant–pathogen interactions [7,8].

In mycorrhizal symbiosis, evidence for ckRNAi is more limited but still significant. For example, in the ectomycorrhizal relationship between *Eucalyptus grandis* and *Pisolithus microcarpus*, miRNA-mediated cleavage of host defense gene transcripts has been observed. The miRNA Pmic_miR-8, when inhibited, impairs the maintenance of the mycorrhizal association, underscoring the importance of such regulatory mechanisms in the success of these symbioses [40].

In arbuscular mycorrhizal (AM) symbiosis, predictions of sRNA–host transcript interactions based on in silico data suggest the potential for ckRNAi, although experimental validation is still needed. AM fungi, such as *Rhizophagus irregularis*, possess a complete set of RNAi machinery, which is expanded in some species like *R. irregularis* [41]. As in pathogenic and ectomycorrhizal systems, the fungal partner may target both host defense genes and membrane-remodeling phospholipases, which are important for forming the peri-arbuscular membrane (PAM), a crucial site for nutrient exchange in AM symbiosis [12].

### 3.2. Plant-to-Microbial Symbiont Communication

Research on reciprocal communication between plants and their microbial symbionts is less developed compared to plant–pathogen interactions, but there are some promising findings. In silico analyses have suggested that miRNAs from *Populus species* may target genes in ectomycorrhizal fungi such as *Laccaria bicolor* and AM fungi like *R. irregularis* [13]. These predictions remain to be experimentally verified, but they suggest the possibility that plants regulate functions in their fungal partners. Given the extensive bidirectional signaling that underpins the establishment and maintenance of symbioses, it is likely that plants do indeed modulate key aspects of their microbial symbionts’ functions during these interactions [42].

Additionally, although bacterial RNAi machinery is absent in root-nodulating symbioses, evidence from host-induced gene silencing (HIGS) in other systems, such as *Arabidopsis*–*Pseudomonas syringae*, indicates that plant-derived sRNAs can silence bacterial genes through mechanisms that are yet to be fully understood [11]. This opens the possibility for reciprocal ckRNAi in both root-nodulating and mycorrhizal symbioses.

### 3.3. Evidence for RNA Transport in AM Symbiosis

Host-induced gene silencing experiments in AM symbiosis further support the idea that plants may transport RNA molecules to fungi to modulate fungal gene expression. Several studies have shown that sRNAs can silence genes in *Rhizophagus* spp. and *Gigaspora* spp., including transporters, receptors, and effector proteins, through the transfer of sRNAs from plant to fungus [43,44,45]. Moreover, recent transmission electron microscopy (TEM) and tomography studies of the PAM and peri-arbuscular space (PAS) suggest the presence of extracellular vesicles (EVs) in the PAS, potentially indicating RNA transport from the plant to the fungus [46,47]. These findings align with the role of EVs in sRNA transport in pathogenic systems and provide further evidence that reciprocal ckRNAi may occur in AM symbiosis [13]. The fusion of multivesicular bodies with the PAM suggests that some EVs are plant-derived, possibly modulating fungal gene expression through RNA and protein delivery, akin to mechanisms seen in mammalian EVs [39,47].

While research on the role of ckRNAi in plant–microbe mutualistic interactions is still emerging, growing evidence supports the idea that sRNAs play a crucial role in regulating both host and symbiont gene expression. In both root-nodulating and mycorrhizal symbioses, sRNA-mediated gene silencing mechanisms are essential for the successful establishment and maintenance of these partnerships. Additionally, there is increasing support for reciprocal communication, where plants may regulate microbial gene expression through the transport of sRNAs and other RNAs. Further experimental studies, particularly in the context of arbuscular mycorrhizal symbiosis, are needed to fully understand the dynamics of ckRNAi in mutualistic interactions and its potential applications in agricultural biotechnology.

## 4. Challenges and Limitations in Investigating ckRNAi Mechanisms in Plant–Microbe Interactions

In silico predictions on *R. irregularis* sRNA targets in *Medicago* suggest that ckRNAi could occur within this symbiosis [12]. However, this remains speculative until experimentally validated. The challenge lies in demonstrating RNA transfer between the plant and fungus in a natural setting, as the detection methods for RNA in extracellular vesicles are still limited [12]. Experimental validation remains a major hurdle in confirming ckRNAi’s role in mutualism. The difficulty of distinguishing between direct RNA transfer and indirect effects due to other signaling mechanisms complicates interpretation. Techniques used in pathogenic ckRNAi studies, such as profiling extracellular sRNAs and AGO pull-down assays, could be adapted to study AM symbiosis. However, these methods are not without challenges, such as the difficulty of detecting small RNA exchanges in the complex environment of the root–fungus interface. While studies indicate fungal-to-plant transfer of *R. irregularis* sRNAs, confirming the functional role of these sRNAs in modulating host gene expression remains challenging. Additionally, interpreting the biological significance of the transferred RNA requires careful consideration of the experimental context and the potential for non-specific interactions. Confirming target transcript cleavage through 5′ RACE assays is a valuable approach but can be complicated by the difficulty of isolating and validating the precise target sites of fungal sRNAs in the plant. Additionally, ectopic expression of fungal sRNAs may not fully capture the natural regulatory mechanisms involved in symbiosis. Observing colonization phenotypes is useful, but the interpretation of these phenotypes can be confounded by other microbial interactions or plant responses that are not directly related to RNAi. This raises the need for more specific markers to track sRNA-mediated effects. Phenotyping AM symbiosis in host plants with overexpressed targets of *R. irregularis* sRNAs could be informative; however, the complexity of plant–microbe interactions makes it difficult to isolate the effect of sRNAs alone. Additionally, compensatory mechanisms in the plant could mask the effects of specific sRNAs. Understanding reciprocal plant-to-fungus ckRNAi is equally challenging. Experimental validation of plant-to-fungus RNA transfer is hindered by the difficulty of detecting and quantifying plant-derived sRNAs in fungal cells, and the lack of efficient transformation systems for *R. irregularis* limits progress in this area.

Recent in silico predictions suggest that *Populus* spp. miRNAs could target *R. irregularis* transcripts through ckRNAi, though no symbiotically relevant genes have been confirmed yet. The reliance on bioinformatics predictions introduces potential false positives, and experimental validation remains critical. Furthermore, it is unclear whether these predicted interactions have a significant biological impact. A stringent bioinformatics pipeline is essential, but bioinformatics predictions must be interpreted cautiously. RNA profiles upregulated during symbiosis may not always reflect direct sRNA–mRNA interactions, and differentiating between direct and indirect effects remains a significant challenge in the field [17,19]. Confirming these interactions experimentally remains challenging due to the difficulty of distinguishing true sRNA–mRNA interactions from background noise. Although RACE assays are useful, they may not always provide conclusive evidence due to the complexity of plant–microbe interactions and the potential for non-specific RNA binding. Strengthening confirmation through AGO association studies is promising but can be complicated by the need for highly specific detection methods. AGO binding alone does not always guarantee functional RNAi activity, and further studies are needed to validate the biological relevance of sRNA–AGO interactions in a plant–microbe context [19].

The inability to transform *R. irregularis* poses a significant limitation for directly studying the role of fungal sRNAs in ckRNAi. This gap in research tools hampers the ability to perform targeted functional studies and necessitates reliance on indirect methods, such as host-induced gene silencing (HIGS), which may not fully replicate the natural processes of RNA transfer. HIGS is a promising alternative for assessing fungal gene function, but it may not capture the full complexity of sRNA-mediated regulation in symbiosis. Additionally, using HIGS to silence or overexpress fungal target genes may affect the broader microbial community in ways that complicate the interpretation of results. While these steps are essential for proving ckRNAi’s presence, they may not be sufficient to conclusively demonstrate its significance in AM symbiosis. The development of more refined tools for RNA transfer studies, including improved transformation methods for fungi, will be critical for advancing the field. Further experiments modulating sRNA types and abundance could explore manipulating this communication mechanism. However, altering sRNA abundance or types could have unintended consequences on the overall plant–microbe relationship, and these experiments need to be carefully controlled to avoid misleading conclusions.

## 5. Future Directions and Perspectives

Currently, there is a lack of conclusive in vivo evidence supporting the involvement of cross-kingdom RNA interference (ckRNAi) in regulating arbuscular mycorrhizal (AM) symbiosis. However, existing research strongly suggests that sRNAs play a key role in regulating plant interactions with both microbial mutualists and pathogens. Predictions based on in silico models indicate that AM fungi might utilize ckRNAi to influence host defense mechanisms and regulate arbuscule development by modulating host membrane remodeling [12]. Since AM symbiosis facilitates the exchange of inorganic minerals for fatty acids, it is possible that the fungus might also target host processes that alter the balance of this exchange in its favor. For instance, it could regulate the plant’s fatty acid transport, biosynthesis, or phosphate metabolism, potentially shifting the host’s needs to benefit the fungus. Such mechanisms might resemble the natural regulation of phosphate-related genes through RNAi [48]. These hypotheses remain to be confirmed through experimental studies.

The reciprocal nature of this communication, however, has not yet been experimentally demonstrated. Despite this, the bidirectional signaling processes crucial for establishing and maintaining AM symbiosis suggest that plant-to-fungus ckRNAi is not only possible but likely. Early in silico findings support this, suggesting that plants might regulate the metabolism of AM fungi, controlling functions like phosphate or lipid uptake and arbuscule development timing to balance the phosphate–lipid exchange. Moreover, ckRNAi might be essential for managing the development of non-self organisms, as transmission electron microscopy (TEM) studies have identified membrane tubule formation in the paramural space, a feature that is similar to what is observed in the invasive hyphal growth of *Ustilago maydis* [38,47]. This may point to an aggressive fungal invasion that the host may need to regulate through mechanisms like ckRNAi, which have been shown to control similar processes in plant–pathogen interactions [23,24].

Recent discoveries in pathogen–plant ckRNAi offer new avenues for exploration. For example, the loading of endosymbiont sRNAs into host AGO4 proteins presents an opportunity for transcriptional gene silencing, potentially enabling long-term regulation of gene expression through DNA methylation. Such regulation could provide benefits to mutualistic relationships by stabilizing interactions between symbiotic partners. The role of DNA methylation in symbiosis has been highlighted in ectomycorrhizal relationships, where hypomethylation in *Populus* plants correlates with reduced association with the *Laccaria bicolor* fungus [49]. Exploring whether similar DNA methylation mechanisms occur in AM symbiosis could offer new ways to enhance symbiotic stability, particularly in agricultural crops.

Additionally, the recently observed transfer of mRNAs between host plants and pathogenic fungi [20,50] may also apply to mutualistic symbioses, suggesting that such mechanisms could play crucial roles in both pathogenicity and mutualistic colonization. However, it remains possible that this transfer is specific to microbes lacking traditional RNAi machinery, which would complement their inability to engage in ckRNAi. Verifying this hypothesis will be essential for broadening our understanding of communication and regulation in AM symbiosis and could lead to strategies to optimize these symbiotic relationships for agricultural benefit.

## Figures and Tables

**Figure 1 ijms-26-02631-f001:**
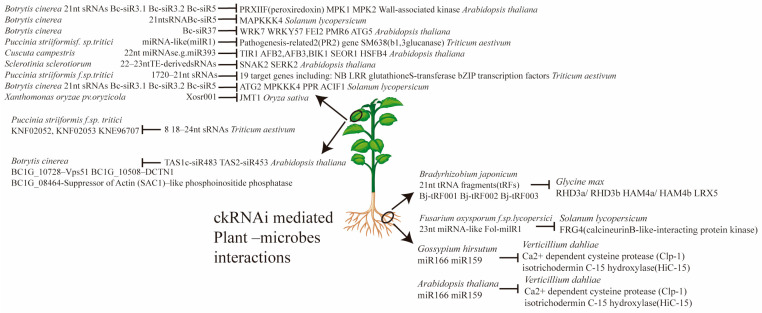
Cross-kingdom RNAi-mediated plant–microbe interactions in leaves and roots of plant. Various microbes interact with plants, resulting in alterations of miRNA levels which are responsible for regulating the expression level of respective target genes.

## Data Availability

No new data were created or analyzed in this study.

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
