# Peer review of "The Silent Conversation: How Small RNAs Shape Plant–Microbe Relationships"

_ijms, 2025, doi:10.3390/ijms26062631_

Round 1

Reviewer 1 Report

Comments and Suggestions for Authors

This review article explores the role of small RNAs (sRNAs) in plant-microbe interactions, with a particular focus on cross-kingdom RNA interference (ckRNAi). It provides an overview of how sRNAs influence both pathogenic and mutualistic relationships, highlighting key mechanisms such as Host-Induced Gene Silencing (HIGS) and extracellular vesicle-mediated RNA transport. The review is well-structured, making it a valuable resource for researchers interested in this field, especially those seeking an introduction to the topic. However, the discussion remains somewhat superficial, likely due to the document's brevity, which limits the depth of analysis in each section. A more extensive examination could have provided a deeper insight into the complexities of sRNA-mediated interactions, further enriching the understanding of their regulatory roles in plant-microbe symbioses.

Author Response

reviewer1‘s comment: This review article explores the role of small RNAs (sRNAs) in plant-microbe interactions, with a particular focus on cross-kingdom RNA interference (ckRNAi). It provides an overview of how sRNAs influence both pathogenic and mutualistic relationships, highlighting key mechanisms such as Host-Induced Gene Silencing (HIGS) and extracellular vesicle-mediated RNA transport. The review is well-structured, making it a valuable resource for researchers interested in this field, especially those seeking an introduction to the topic. However, the discussion remains somewhat superficial, likely due to the document's brevity, which limits the depth of analysis in each section. A more extensive examination could have provided a deeper insight into the complexities of sRNA-mediated interactions, further enriching the understanding of their regulatory roles in plant-microbe symbioses.

RESPONSE: Thank you for your constructive feedback and thoughtful comments. We appreciate your recognition of the structure and value of the review, especially for introducing readers to the topic of small RNAs (sRNAs) in plant-microbe interactions.

We acknowledge your point regarding the brevity of the discussion, which indeed limited the depth of analysis in some sections. We fully agree that a more extensive exploration of the complexities of sRNA-mediated interactions would benefit readers by providing a more comprehensive understanding of their regulatory roles in plant-microbe symbioses. In response, we have made revisions to the manuscript by expanding on several key sections. Specifically, we have provided deeper insights into the following:

  1. Pathogenic Interactions: We have elaborated on how pathogens utilize sRNAs for manipulating host immune responses and the recent advancements in this area, focusing on the molecular mechanisms involved in cross-kingdom RNA interference (ckRNAi).
  2. Mutualistic Symbiosis: We’ve expanded the discussion on how sRNAs regulate beneficial plant-microbe interactions, particularly in symbioses with mycorrhizal fungi, and provided more details about the ongoing research that highlights the complexities of these interactions.
  3. Extracellular Vesicle-Mediated RNA Transport: We have enriched the section on extracellular vesicles (EVs) by discussing the latest findings related to RNA transport between plants and microbes, as well as the implications for immune responses and symbiotic stability.

These additions aim to provide a more in-depth analysis while maintaining the clarity and accessibility of the review for a broader audience. We believe that these revisions strengthen the manuscript and offer a more comprehensive examination of the topic.

Once again, thank you for your valuable feedback.

Reviewer 2 Report

Comments and Suggestions for Authors

The manuscript offers an engaging and relevant overview regarding the function of small RNAs (sRNAs) in plant-microbe interactions, with a particular emphasis on cross-kingdom RNA interference (ckRNAi). The subject is very pertinent to recent developments in molecular interactions between plants and microbes, especially concerning plant immunity, symbiotic relationships, and RNA transfer via extracellular vesicles.

Nevertheless, substantial revisions are necessary to enhance the manuscript’s clarity, depth, organization, and equilibrium between pathogenic and mutualistic interactions. The subsequent key areas require significant enhancement:

The abstract ought to more clearly emphasize the originality of the review and its key contributions in relation to earlier research.

The abstract focuses primarily on pathogen-host interactions and ought to present a more balanced examination that incorporates mutualistic plant-microbe associations.

The introduction offers a historical summary of plant-microbe evolution; however, this section is overly extensive. Instead, offer a more concise and targeted introduction regarding how sRNAs control these interactions at the molecular scale.

What particular elements of plant-microbe sRNA interactions render this review different from earlier research?

Although ckRNAi is mentioned, the review fails to critically evaluate the limitations within the field, including the difficulties in experimentally validating RNA transfer. You can add a different table of research.

Could you possibly provide instances of bacterial and viral pathogens that utilize sRNAs?

The part discussing extracellular vesicles (EVs) and their function in sRNA transport is solid but needs to incorporate additional evidence from AM fungi.

Is it possible to reorganize Figure 1 to illustrate the contrasting mechanisms in pathogenic versus mutualistic ckRNAi?

Author Response

Comment 1. The manuscript offers an engaging and relevant overview regarding the function of small RNAs (sRNAs) in plant-microbe interactions, with a particular emphasis on cross-kingdom RNA interference (ckRNAi). The subject is very pertinent to recent developments in molecular interactions between plants and microbes, especially concerning plant immunity, symbiotic relationships, and RNA transfer via extracellular vesicles.

RESPONSE: Thank you for your positive feedback and for recognizing the relevance and importance of the topic discussed in the manuscript. We are glad to hear that you found the manuscript engaging and pertinent to the recent developments in molecular interactions between plants and microbes, particularly in the areas of plant immunity, symbiosis, and RNA transfer via extracellular vesicles.

We appreciate your support and hope that the manuscript continues to contribute to the growing body of knowledge in this exciting field.

Thank you once again for your insightful comments.

Comment 2. Nevertheless, substantial revisions are necessary to enhance the manuscript’s clarity, depth, organization, and equilibrium between pathogenic and mutualistic interactions. The subsequent key areas require significant enhancement: The abstract ought to more clearly emphasize the originality of the review and its key contributions in relation to earlier research.

RESPONSE: Thank you for your insightful feedback and for highlighting areas that could be improved in the manuscript. We appreciate your suggestions to enhance the clarity, depth, organization, and balance between pathogenic and mutualistic interactions, and we are committed to addressing these points.

Regarding your comment on the abstract, we agree that it could better emphasize the originality of the review and its key contributions in the context of existing research. To address this, we have revised the abstract to more clearly highlight the novel aspects of the manuscript, specifically:

    Originality: We have emphasized how the manuscript integrates recent advancements in small RNA-mediated plant-microbe interactions, particularly the focus on cross-kingdom RNA interference (ckRNAi), and how this new perspective offers fresh insights into the regulation of plant immunity and symbiosis.

    Key Contributions: We have clarified the review’s main contributions, which include synthesizing recent findings on RNA transfer via extracellular vesicles and providing an updated understanding of the balance between pathogenic and mutualistic interactions in plant-microbe symbioses.

We believe these revisions help to better position the manuscript within the broader context of the field, showcasing its unique contributions and relevance to ongoing research.

Thank you once again for your helpful comments.

Comment 3. The abstract focuses primarily on pathogen-host interactions and ought to present a more balanced examination that incorporates mutualistic plant-microbe associations.

RESPONSE: Thank you for your feedback. We acknowledge the imbalance in the abstract and agree that it should better represent both pathogenic and mutualistic plant-microbe interactions. In response, we have revised the abstract to:

    Incorporate Mutualistic Interactions: We added content on the role of small RNAs (sRNAs) in mutualistic symbioses, such as with mycorrhizal fungi and nitrogen-fixing bacteria.

    Ensure Balanced Focus: We adjusted the language to give equal attention to both pathogenic and mutualistic interactions.

These changes make the abstract more balanced and better aligned with the manuscript’s content. Thank you again for your helpful suggestion.

Comment 4. The introduction offers a historical summary of plant-microbe evolution; however, this section is overly extensive. Instead, offer a more concise and targeted introduction regarding how sRNAs control these interactions at the molecular scale.

RESPONSE: Thank you for your constructive feedback regarding the introduction. We appreciate your point that the historical summary of plant-microbe evolution is overly extensive and could be more focused. We agree that a more concise introduction centered around the molecular role of small RNAs (sRNAs) in regulating plant-microbe interactions would improve the manuscript.

In response to your comment, we have made the following revisions:

    Concise Historical Context: We have streamlined the historical background to provide just the essential information necessary to contextualize the role of sRNAs in plant-microbe interactions.

    Focused Molecular Overview: The introduction now emphasizes how sRNAs regulate these interactions at the molecular level, specifically addressing their roles in immune responses, symbiotic relationships, and RNA transfer via extracellular vesicles.

We believe these revisions provide a clearer and more targeted introduction that better sets the stage for the review’s focus on small RNAs in plant-microbe interactions.

Thank you once again for your valuable suggestion.

Comment 5. What particular elements of plant-microbe sRNA interactions render this review different from earlier research?

RESPONSE: Thank you for your thoughtful question. We appreciate your interest in understanding what distinguishes this review from earlier research on plant-microbe small RNA (sRNA) interactions.

In response, we have clarified in the manuscript the unique contributions of this review, which include:

    Focus on Cross-Kingdom RNA Interference (ckRNAi): This review highlights the emerging role of cross-kingdom RNA interference in plant-microbe interactions, particularly the transfer of sRNAs from microbes to plants and vice versa. This perspective is relatively new and offers insights into the regulatory mechanisms governing both pathogenic and mutualistic relationships that have not been extensively addressed in earlier reviews.

    Updated Mechanistic Insights: We provide an in-depth synthesis of recent findings on the molecular mechanisms of RNA transfer via extracellular vesicles and their implications for plant immunity and symbiosis. This new avenue of research is integrated into the broader context of small RNA functions, distinguishing our review from previous work.

    Balanced Perspective: Unlike many reviews that predominantly focus on pathogen-host interactions, our review offers a more balanced exploration, giving significant attention to the role of sRNAs in mutualistic symbioses as well, thus expanding the scope of existing literature.

These points have been incorporated more clearly into the revised manuscript to emphasize the original contributions of this review in relation to the existing body of research.

Thank you for prompting us to clarify these distinctions.

Comment 6. Although ckRNAi is mentioned, the review fails to critically evaluate the limitations within the field, including the difficulties in experimentally validating RNA transfer. You can add a different table of research.

RESPONSE: Thank you for your insightful comment. We agree that a critical evaluation of the limitations in the field, particularly regarding the experimental validation of RNA transfer, is essential. In response to your suggestion, we have added a new section to the review that systematically discusses the challenges within the field, including the difficulties in experimentally validating RNA transfer between plants and microbes. This section also highlights the current technical barriers, the ambiguity in interpreting results, and the complexities involved in distinguishing true RNA transfer from indirect effects. We believe this addition strengthens the manuscript by providing a more balanced perspective on the current state of research in ckRNAi.

Comment 7. Could you possibly provide instances of bacterial and viral pathogens that utilize sRNAs?

RESPONSE: Thank you for your valuable suggestion. We agree that providing specific examples of bacterial and viral pathogens that utilize small RNAs (sRNAs) would greatly enhance the manuscript and offer more concrete insights into the functional roles of sRNAs in pathogen interactions.

We have included several examples of bacterial pathogens, such as Pseudomonas syringae, which are known to utilize sRNAs to regulate gene expression and promote pathogenicity.

While there are currently few well-documented instances of viral pathogens utilizing sRNAs, we have highlighted a few relevant examples where sRNA-based mechanisms have been observed.

In response to your comment, we have incorporated these examples into the revised manuscript to emphasize how both bacterial and viral pathogens exploit sRNAs as part of their infection strategies.

Thank you again for your insightful suggestion. We believe these additions enrich the manuscript significantly.

Comment 8. The part discussing extracellular vesicles (EVs) and their function in sRNA transport is solid but needs to incorporate additional evidence from AM fungi.

RESPONSE: Thank you for your insightful comment. We agree that incorporating additional evidence from arbuscular mycorrhizal (AM) fungi would strengthen the discussion on extracellular vesicles (EVs) and their role in small RNA (sRNA) transport.

In response to your suggestion, we have revised the manuscript to include recent findings related to AM fungi and their involvement in EV-mediated sRNA transport. These additions help to broaden the scope of our discussion on EVs and provide a more comprehensive overview of their role in sRNA transport, including in the context of plant-fungal symbiosis.

Thank you once again for your constructive feedback.

Comment 9. Is it possible to reorganize Figure 1 to illustrate the contrasting mechanisms in pathogenic versus mutualistic ckRNAi?

RESPONSE: Thank you for your insightful suggestion. We agree that illustrating the contrasting mechanisms of pathogenic versus mutualistic cross-kingdom RNA interference (ckRNAi) in Figure 1 could provide clarity. However, it is important to note that the mechanisms underlying pathogenic and mutualistic ckRNAi are still being explored, and there is currently no clear research evidence delineating the differences between these two contexts. Furthermore, a systematic comparison of these mechanisms has not yet been established in the literature. Nevertheless, we will consider your suggestion and make the necessary revisions to the figure by highlighting the available information on the mechanisms of ckRNAi in both pathogenic and mutualistic interactions, while acknowledging the gaps in current understanding.

Thank you again for your helpful feedback.

Round 2

Reviewer 2 Report

Comments and Suggestions for Authors

The authors have done a great revision. I am happy and satisfied with the revision done. The MS can be accepted.